# DLGrapher: Dual Latent Diffusion for Attributed Graph Generation

## Abstract

Graphs for applications like social data and financial transactions are particularly complex, with large node counts and high-dimensional features. State-of-the-art diffusion graph synthesizers model the node structure via discrete diffusion and are, unfortunately, limited to small-scale graphs with few to no features. In contrast, continuous diffusion models capture rich node features well, but have issues faithfully modelling connectivity. In this paper, we design DLGrapher, a dual latent diffusion framework for jointly synthesizing large graph structures and high-dimension node features. DLGrapher models node features and structure as a joint latent representation. Structure-wise, we design a reversible coarsening scheme to merge pairs of similar neighboring nodes and their respective edges after encoding node features through a structure-aware variational autoencoder. To capture the dependencies between node features and the graph structure, DLGrapher trains a single diffusion over a dual denoising objective, one for the continuous node representations and another for the discrete edge connectivity. We extensively evaluate DLGrapher's performance on three complex social graph datasets against baselines combining tabular and graph synthesizers. Our solution fares 12.9x better at statistically capturing feature-structure interaction and 25.2% better at downstream tasks thanks to the dual diffusion on average and the latent compressed representation increases throughput by 2.5X. Furthermore, we maintain competitive synthesis quality for simple-featured molecular graphs and structure-only synthetic graphs while drastically reducing computation in the latter case.

## 1 Introduction

Graphs are widely used to model the interactions of social media users (Rozemberczki & Sarkar, 2021), financial transaction (Altman, 2021), and molecules in biology (Wu et al., 2017). Attributed graphs are characterized by their graph structures representing interactions among nodes and node features representing unique characteristics. Figure 1 shows an example of an attributed graph: users with distinct features are nodes, and the connectivity of edges shows their interactions. Node features influence the graph structure, which in turn affect the feature values. Consider an example of a social network with users producing and consuming content; popular creators with many views also tend to have the most people choosing to follow their profile. To date, graphs without attributes are increasingly synthesized by generative models Chen et al. (2023); Bergmeister et al. (2024); Dai et al. (2020) in search for unseen patterns or as an alternative for data sharing solution.

The state-of-the-art graph generative models draw methodologies from generative adversarial networks Martinkus et al. (2022), transformers Vignac et al. (2023), and diffusion Jo et al. (2024), with the main focus on the graph structure. To model the discrete nature of graph structure, the prior work Simonovsky & Komodakis (2018) first applies encoder networks to find graphs' continuous latent representation, which then can be straightforwardly learned and synthesized by a diffusion model in continuous space. The quality of

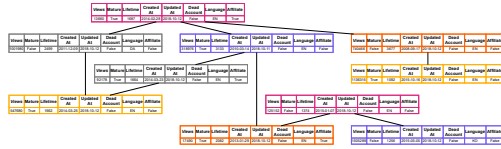

Figure 1: Example subgraph with complex node data from Twitch Gamers dataset Rozemberczki & Sarkar (2021).

synthetic graphs is thus limited by the capacity of the encoder Recently, discrete graph diffusion not only shows a remarkable quality in synthesizing molecular structures by modeling the discrete process of edge connectivity, but also captures the node features through single conditioning, e.g., molecular structure with certain properties Vignac et al. (2023). However, such discrete models are limited in synthesizing either large graph structures or graphs with complex features. While the discrete diffusion model well captures the connectivity among nodes, it does not scale to large numbers of nodes The maximum number of conditions that can be handled by the prior art is two because of the exponentially growing complexity of cross-features correlation.

In this paper, we propose DLGrapher, a **D**ual **L**atent **Graph** diffusion model, which is capable of learning from large and complex attributed graphs and efficiently synthesizing graphs with rich features. DLGrapher aims to combine the advantage of the scalability of latent diffusion and the graph quality of discrete diffusion models. The core design features of DLGrapherare the structure-aware latent representations of attributed graphs and the dual diffusion model, which jointly de-noise the discrete latent of the structure and continuous latent of the features. DLGrapherfirst models the feature and structure through a structure-aware feature encoder-decoder networks and a novel reversible coarsening scheme, respectively. When searching for the features embeddings, we include the edge connectivity into the variational encoder networks. The coarsening scheme merges pairs of similar neighboring nodes and their edges, concatenating their node features. To capture the dependency between structure and features, the dual diffusion model of DLGraphercombines the training losses from discrete diffusion on the structure latent and from the continuous diffusion on the feature latent and then uses the combined loss to train the respective denoising processes of the structure and feature. We evaluate DLGrapheragainst the state-of-the-art graph and tabular diffusion models, in terms of their graph structure metrics, feature quality metrics, inter-dependency between structure and features, and downstream tasks performance. In small-scale attributed graphs, DLGrapher outperforms the baseline in all four types of metrics, capturing feature-structure interaction 12.9x better and improving 25.2% better at down-stream tasks.

The novel contributions of DLGrapherare the following: (i) the first-of-kind generative model for attributed graphs, complex in structure and rich in feature, (ii) a compact and structure-ware joint representation of structure and features, (iii) a dual latent diffusion framework that jointly optimizes the synthesis of discrete latent structure and continuous latent of features, and (iv) evaluation on attributed graphs of different sizes in social networks, and molecular biology.

## 2 RELATED WORK

In recent years, **diffusion models** have been at the forefront of research into synthetic data generation over a multitude of modalities, like images (Ho et al., 2020), audio (Liu et al., 2024), video (Ho et al., 2022), tabular data (Kotelnikov et al., 2023; Zhang et al., 2024), and even discrete settings like language modeling (Lou et al., 2024). Latent formulations of such models learn over a lower dimension encoded version of the input data and have been shown to help reduce computation requirements and even improve synthesis in image (Rombach et al., 2022) and tabular (Shankar et al., 2024) contexts.

The two main **graph generation architectures** are based on autoregressive and diffusion approaches, with the latter offering higher sample quality with generally increased overhead. Within diffusion, a further differentiator is the graph noising model, which can be continuous, as in most other modalities, or discrete, better matching the nature of graph structures. Discrete noising ensures that noisy representations remain valid graphs and can better maintain sparsity during synthesis. As for latent graph diffusion variants, current efforts lie in 3D molecule generation, which strictly focuses on modeling the Euclidean coordinates and properties of atoms (Xu et al., 2023; You et al., 2024). For the restricted case of unattributed graphs, recent autoregressive methods, like Dai et al. (2020) and Karami (2024), harness the sparsity of graphs or hierarchical structures to model connectivity, respectively. From diffusion approaches, the discrete Chen et al. (2023) improves efficiency by denoising part of the structure at a time. In contrast, Bergmeister et al. (2024) expands nodes at every denoising step to generate graphs with up to thousands of nodes. Although such models may be augmented to incorporate a distinct process for node or edge attributes, Jo et al. (2022) shows that simultaneously generating structure and features leads to considerably better results. For attributed generators, existing models generate much smaller graphs due to the increased problem complexity,

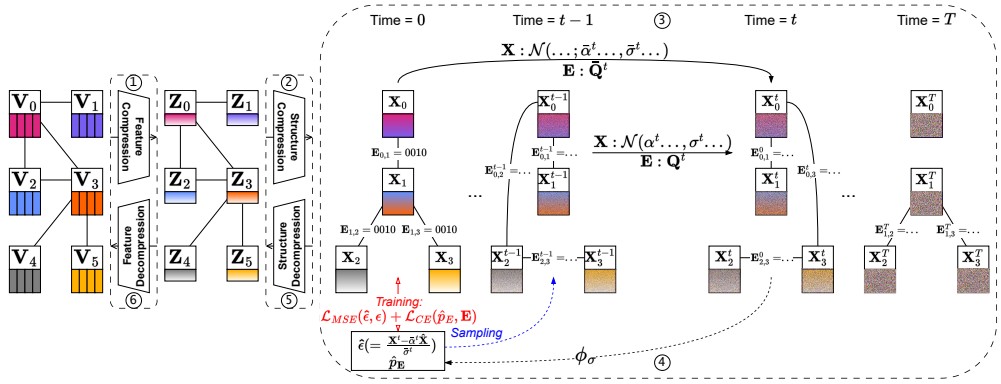

Figure 2: DLGrapher detailed overview: (i) encoding feature latent, (ii) encoding structure latent, (iii) forward process of dual diffusion, (iv) denoising process of dual diffusion, (v) decoding structure latent, and (iv) decoding feature latent.

even if most only integrate single-class nodes or edges. These include Jang et al. (2024), a hierarchical autoregressive model, Kong et al. (2023), which performs autoregressive diffusion, and Vignac et al. (2023), which proposes a discrete denoising model that predicts individual nodes and edges to generate graphs of under 200 nodes. In Jo et al. (2024), authors propose a graph mixture diffusion model that predicts graph mixture focusing on the global graph structure, additionally allowing the synthesis of simple continuous data for a node alongside its class. DLGrapher is the first model to handle complex node representations with many features of different types, like those of tabular data, while increasing the size of generated attributed graphs at similar computation costs.

**Coarsening** is a technique for reducing dimensionality when working with large graphs while preserving key properties. Many versions consist of fixed algorithms, but newer works explore variants learnable through neural networks as well (Cai et al., 2021). All such methods operate on the graph structure, for example, striving to preserve similar spectral properties (Jin et al., 2020), but some methods also account for node features (Kumar et al., 2023). Unlike prior art, which is not designed to recover the original graph from the reduced graph, our proposed coarsening scheme is reversible.

## 3 DLGRAPHER: DUAL LATENT GRAPH DIFFUSION MODEL

This section describes DLGrapher, which tackles the generation of graphs with high-dimension node features. The DLGrapherframework combines two main components: a structure-aware latent encoding mechanism and a dual diffusion backbone. DLGrapher first represents the attributed graphs into the discrete structure embedding through a reversible coarsening scheme and continuous feature embedding through a structure-aware feature encoder. The structure embedding is still a valid graph with aggregated virtual nodes and edges, hence applicable for high quality discrete graph diffusion. The compact embedding reduces overhead and enhances the generation capability with respect to the graph size and feature complexity. The dual denoising diffusion model enables not only to synthesize complex node features and an accurate connectivity structure, but also, importantly, to capture their interdependencies.

To synthesize attributed graph shown in Figure 2, DLGrapher is composed of three components. (i) Structure-aware feature encoding-decoding networks. These can encode node features into continuous latent embeddings in a structure-aware manner and decode the latent back to the feature space. (ii) Reversible structure coarsening scheme. It finds the structure embedding as a lower dimension graph, i.e., virtual nodes and edges aggregated from the original nodes and edges, through coarsening the structure based on neighboring node pairs. (iii) Dual diffusion model. It learns to synthesize the joint embedding of an attributed graph - a lower dimension graph with a feature embedding, through continuous and discrete (de)noising processes on the feature and structure embeddings. The feature encoder and dual diffusion model are transformer networks whose parameters are learned through the training data of attributed graphs. In contrast, the reversible coarsening scheme is a fixed bidirectional transformation function.

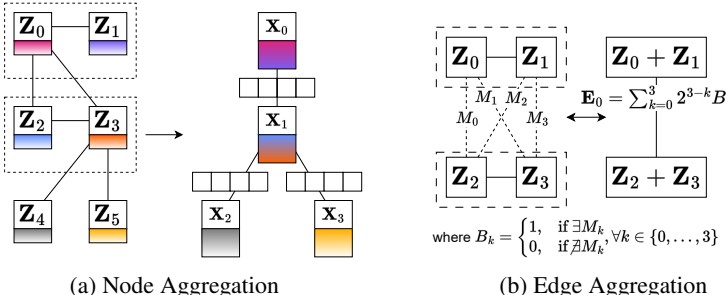

(a) Node Aggregation

(b) Edge Aggregation

Figure 3: Latent graph through structure coarsening and nodes/edge aggregation. Original graph with latent feature $(\mathbf{Z}, \mathbf{M})$ transformed into latent graph $(\mathbf{X}, \mathbf{E})$. Nodes are reduced from 6 to 4 virtual nodes, and edge are reduced from 6 to 4 virtual edges associated with different types.

Subsequently, for generating synthetic attributed graphs, we denoise a random graph from the latent space, i.e., a virtual graph with latent features, apply reverse coarsening to restore the original space of nodes/edges, and decode node features back to the same dimension as the original data.

**Notations and Definitions**: The original graphs $G_o = (\mathbf{V}, \mathbf{M})$ with nodes $\mathbf{V} \in \mathbb{R}^{n \times f}$ and edges $\mathbf{M} \in \{0, 1\}^{n \times n}$, where $n$ is the number of real nodes, and $f$ is the real feature dimension. The original graphs are the training inputs to extract feature embedding, represented as $\mathbf{Z} \in n \times f'$, through the proposed structure-ware VAE, where $f'$ is the dimension of node feature embedding. Then, the coarsening scheme further generates the latent graph embedding, $G = (\mathbf{X} \in \mathbb{R}^{n' \times 2f'}, \mathbf{E} \in \mathbb{N}^{n' \times n'})$ from $(\mathbf{Z}, \mathbf{M})$, where $n'$ is the number of virtual aggregated nodes. The training inputs to the diffusion backbone considered are thus the graphs $G = (\mathbf{X}, \mathbf{E})$.

### 3.1 Embedding of Attributed Graph

We aim to find a compact embedding for attributed graphs, which is still a valid graph applicable to discrete diffusion on the edge connectivity and with a compact node feature representation for continuous latent diffusion. Our embedding procedure has a two-step process, first operating at the node feature and then at the structural level. Node feature embedding ensures a decreased dimensionality compared to the original data. This also eases the subsequent step of coarsening the graph structure, which needs to aggregate nodes and concatenate their features.

**Structure-aware VAE**: Node features per node of attributed graph are essentially an individual row in the feature table, shown in Figure 1. Representing graphs as adjacency matrix, the edge connectivity in the graph represents the cross row dependency. In addition to capture the cross-attribute dependency, the embeddings of the features need to address two challenges: capturing the row dependency reflected in the edge connectivity and modeling the categorical and continuous node attributes. We design a structure-aware variational autoencoder (sVAE), outputting a latent representation of the node features described by a Gaussian. Specifically, the node feature is reduced by sVAE from $\mathbf{V} \in \mathbb{R}^{n \times f}$ to the latent embedding $\mathbf{Z} \in \mathbb{R}^{n \times f'}$, with $f' < f$. This latent representation is then used in the diffusion process to synthesize the node features.

We design sVAE as a two-layer network of SageConv operators (Hamilton et al., 2017) for both the encoder and decoder; see Appendix D for more details. We choose SageConv as it is a fast convolutional layer that can aggregate information from each node's neighbors, given that the structural embeddings are also obtained through pairs of adjacent nodes. Thus, at each layer, the representation of a node in the graph is updated according to its own current value and that of its graph neighbors. We optimize the encoder-decoder network with a weighted combination of a mean squared error loss targeting reconstruction quality and a KL divergence loss acting as a regularizer, ensuring that the learned latent distribution is similar to some preselected prior, i.e., Gaussian distribution (Kingma & Welling, 2014). Moreover, to cater to the categorical and continuous features, we customize the activation function at the decoder, using softmax for categorical features and sigmoid for continuous ones.

### 3.1.1 REVERSIBLE STRUCTURE COMPRESSION

The objective here is twofold. First, we want to transform the graph into a compact latent that is still a valid graph, such that the discrete diffusion model can be applied. Secondly, we need to ensure that such a coarsening is reversible. To achieve such aims, we opt for a graph to graph transformation algorithm, instead of using a learning approach, which typically represents structures in another space Simonovsky & Komodakis (2018). We coarsen the nodes and the edges between them, by aggregating pairs of similar nodes and their edges - termed here virtual nodes and virtual edges. The challenge is how to keep the information of original nodes and edges, e.g., node's latent feature and edge classes, in the structure embedding. The core idea is related to that of graph coarsening (Cai et al., 2021), but our scheme is made to be reversible, allowing to recover the original graph. We outline the coarsening procedure in Algorithm 1.

**Node coarsening**: We first greedily pair up adjacent node pairs with decreasingly similar feature representations into a new virtual node, concatenating their feature representations into a larger latent. The lower bound for $n'$ is $\frac{n}{2}$, when there is an even number of nodes $n'$, but a complete assignment is often not possible, e.g., when $n$ is odd) or is overly computation expensive. We thus allow nodes to remain unpaired, and merge them with a dummy, zero-filled node. Then we connect new nodes when any of their components were initially connected using the edge coarsening algorithm. In order to support coarsening of the edges between virtual nodes, we introduce the edge type in our latent embedding, i.e., $\mathbf{E} \in \mathbb{N}^{n' \times n'}$.

**Edge coarsening**: As all nodes within a pair are always inherently connected, we discard edges within the same pair from the resulting coarse structure. We keep track of edges

---

**Algorithm 1** Structure Coarsening

**Input**: feature encoder $e_\theta$,
$G_o = (\mathbf{V} \in \mathbb{R}^{n \times f}, \mathbf{M} \in \{0,1\}^{n \times n})$
1: $\mathbf{Z}$, edges $\leftarrow e_\theta(\mathbf{V}, \mathbf{M})$, $[(i,j) \,|\, \mathbf{M}_{i,j} = 1]$
2: **for** $(i,j) \leftarrow$ edges sorted by $\min ||\mathbf{z}_i - \mathbf{z}_j||^2$ **do**
3:      **if** $\text{Pair}_i = \varnothing \wedge \text{Pair}_j = \varnothing$ **then**
4:          $\text{Pair}_i \leftarrow \text{Pair}_j \leftarrow \max(\text{Pair}) + 1$
5:          $\mathbf{X} \leftarrow Append(\mathbf{X}, Concat(\mathbf{Z}_i, \mathbf{Z}_j))$
6:      **else if** $\text{Pair}_i \neq \varnothing \vee \text{Pair}_j \neq \varnothing$ **then**
7:          intraEdges $\leftarrow$ intraEdges $\cup \{(i,j)\}$
8: **for** $i \leftarrow [k \,|\, Pair_k = \varnothing, 0 \leq k < n]$ **do**
9:      $\text{Pair}_i \leftarrow \max(\text{Pair}) + 1$
10:      $\mathbf{X} \leftarrow Append(\mathbf{X}, Concat(\mathbf{Z}_i, \mathbf{O}^{|\mathbf{Z}_i|}))$
11: **for** $(i,j) \leftarrow$ intraEdges **do**
12:      $l, h \leftarrow \min(\text{Pair}_i, \text{Pair}_j), \max(\text{Pair}_i, \text{Pair}_j)$
13:      $\mathbf{E}_{l,h} \leftarrow \mathbf{E}_{l,h} + EncEdge(\text{Pair}, i, j)$ ▷ fig. 3b
14: **return** $\mathbf{X}, \mathbf{E}$

---

in the original graph joining nodes from different pairs (represented in Algorithm 1 by the intraEdges variable). We later aggregate intraEdges in the coarse graph such that any of the four possible edges between two node pairs becomes a single multi-class edge, with each possible class representing a combination of the initial edges. In Algorithm 1, *EncEdge* maps each original edge to a class in the corresponding edge of the coarse graph. Figure 3b visually describes this mapping. Overall we need 16 classes which represent the four possible edges between two nodes in different pairs. Additionally, we symmetrize the resulting adjacency matrix to ensure that the graph remains undirected. Figure 3 presents an example of coarsening nodes and edges. Finally, structural coarsening reduces $\mathbf{Z} \in \mathbb{R}^{n \times f'}$ to $\mathbf{X} \in \mathbb{R}^{n' \times 2f'}$, and $\mathbf{M} \in \{0,1\}^{n \times n}$ to $\mathbf{E} \in \mathbb{N}^{n \times n'}$, with $n' < n$.

**Decoarsening**: We first split back each node representation into two nodes and add back the edges between them. We subsequently expand each edge in the compressed graph to the original graph edges it aggregates, adding them to the new graph structure. Finally, we remove any dummy zero-filled nodes, reindexing the graph to account for any reduction in nodes, and performing a forward pass through the decoder to restore the original feature space of nodes. Algorithm 4 in Appendix describes the decoarsening in more detail.

### 3.2 DUAL LATENT DIFFUSION MODEL

We design a dual diffusion process that predicts individual node features and edge connectivities of **latent graphs**, $G_o = (\mathbf{X}, \mathbf{E})$, using both continuous (de)noising and discrete (de)noising. We follow the framework of DDPM Ho et al. (2020), and aim to find a model $\phi$ parameterized by $\theta$ to synthesize new graphs starting from a noisy latent representation $G^T$ and denoising it over $T \in \mathbb{N}$ steps. A forward noising process defines a set of predetermined probability distributions $q(G^t|G^{t-1})$, such that after $T$ applications starting from a clean graph $G^0$, the resulting representation follows the

Gaussian distribution, independent of the starting graph. For the reverse process, $q(G^{t-1}|G^t)$ gets approximated through the $\hat{p}(G^0|G^t)$ predicted by $\phi_\theta$ and the known $q(G^{t-1}|G^t, G^0)$. For synthesis, a random graph is sampled from some prior, and the model iteratively refines its prediction of the clean version over $t \leftarrow T, ..., 1$ steps. We note that the number of real nodes is assumed fixed and so is the number of virtual nodes after coarsening.

Different from non-attributed graph synthesis, our latent graph needs to capture not only the edge connectivity but also the latent features of each virtual node. We combine them into a single diffusion framework, consisting of continuous diffusion for the feature latent and discrete diffusion for the edge connectivity. The former predicts the value of the latent features, where the latter predicts the virtual edge and their types, i.e., different kinds of connectivity, which are crucial for decoarsening the graph.

**Continuous Feature Latent X** We model the feature latent by adding Gaussian noise with parameters of $\alpha_t$ and variance $\sigma_t$. The forward process is assumed $q(\mathbf{X}^t|\mathbf{X}^{t-1}) = \mathcal{N}(\alpha^t \mathbf{X}^{t-1}, \sigma^{t^2}\mathbf{I})$. With further algebraic substitution, we can write the forward process as $q(\mathbf{X}^t|\mathbf{X}^0) = \mathcal{N}(\mathbf{X}^t; \bar{\alpha}^t\mathbf{X}^0, \bar{\sigma}^t\mathbf{I})$ where $\bar{\alpha}^t = \prod_{i=1}^{t}\alpha_i$ and $\bar{\sigma}^t = \prod_{i=1}^{t}\sigma_i$. For the reverse process, we need to find a denoising model that is able to minimize the mean squared errors, $||\hat{\epsilon} - \epsilon||^2$, between the added Gaussian noise, $\epsilon$, and predicted noise, $\hat{\epsilon}$, outputted from the denoising model.

**Discrete Edge Latent** We directly predict and sample from the distribution of edge types given by the denoised graph, such that the discrete noising produces a valid graph structure after every step Vignac et al. (2023); Chen et al. (2023). To build such a discrete diffusion, we rely on a transition matrix $\mathbf{Q}^t$ that dictates the probability of each edge type jumping to another, based on the prior probability of each edge type. The forward noising process is thus defined:

$$q(\mathbf{E}^t|\mathbf{E}^{t-1})\mathbf{E}^{t-1}\mathbf{Q}^t, q(\mathbf{E}^t|\mathbf{E}^0) = \mathbf{E}^0\bar{\mathbf{Q}}^t \text{ where } \bar{\mathbf{Q}}^t = \prod_{i=1}^{t}\mathbf{Q}_i$$

To solve the denoising process, one needs to find a denoising model that can predict the edge type probability, $\hat{p}_E$, after any number of transition steps. To solve the denoising model of these two latents jointly, we use the graph transformer Vignac et al. (2023) as the model backbone and both latents as inputs due to its attention mechanism to effectively correlate the inputs. The model outputs are the predicted noise for the latent features and the predicted edge types for any given time step. We thus set the training objective to minimize their weighted joint loss of mean square error from the feature noise and cross entropy loss from the edge types weighted by $\lambda$:

$$L((\hat{\epsilon}; \hat{p}_E), (\epsilon; \mathbf{E})) = ||\hat{\epsilon} - \epsilon||^2 + \lambda CrossEntropy(\hat{p}_E, \mathbf{E})$$

| **Algorithm 2** Dual Diffusion Training Step | **Algorithm 3** Dual Diffusion Sampling |
|---|---|
| **Input**: denoising model $\phi_\theta$, $G = (\mathbf{X} \in \mathbb{R}^{n' \times 2f'}, \mathbf{E} \in \mathbb{N}_{15}^{n' \times n'})$ | **Input**: denoising model $\phi_\theta$ |
| 1: $t, \epsilon \sim \mathcal{U}(1, ..., T), \mathcal{N}(\mathbf{O}_n, \mathbf{I}_n)$ | 1: $\epsilon, \mathbf{E}^t \sim \mathcal{N}(\mathbf{O}_n, \mathbf{I}_n), q_E(n)$ |
| 2: $\mathbf{X}^t \leftarrow \bar{\alpha}^t(\mathbf{X}) + \bar{\sigma}^t(\epsilon)$ | 2: **for** $t = T, ..., 1$ **do** |
| 3: $\mathbf{E}^t \sim \mathbf{E}\overline{\mathbf{Q}}^t$ | 3: $\quad$ $\mathbf{f} \leftarrow ExtraFeats(\mathbf{E}^t, t)$ |
| 4: $\mathbf{f} \leftarrow ExtraFeats(\mathbf{E}^t, t)$ | 4: $\quad$ $\hat{\epsilon}, \hat{p}_E \leftarrow \phi_\theta(\mathbf{X}^t, \mathbf{E}^t, \mathbf{f})$ |
| 5: $\hat{\epsilon}, \hat{p}_E \leftarrow \phi_\theta(\mathbf{X}^t, \mathbf{E}^t, \mathbf{f})$ | 5: $\quad$ $\epsilon \sim \mathcal{N}(0, \mathbf{I}_n)$ |
| 6: Opt $||\hat{\epsilon} - \epsilon||^2 + \lambda CrossEntropy(\hat{p}_E, \mathbf{E})$ | 6: $\quad$ $\mathbf{X}^{t-1} \leftarrow \frac{1}{\alpha^t}\mathbf{X}^t - \frac{\sigma^{t^2}}{\alpha^t\bar{\sigma}^t}\hat{\epsilon} + \sigma^{t \rightarrow t-1}\epsilon$ |
| | 7: $\quad$ **for** $(i, j) \leftarrow (1, ..., n) \times (1, ..., n)$ **do** |
| | 8: $\quad\quad$ $\mathbf{E}_{ij}^{t-1} \sim \sum_e q(e_{ij}^{t-1} \mid e_{ij} = e, e_{ij}^t)\hat{p}_{E_{ij}}(e)$ |
| | 9: **return** $(\mathbf{X}^0, \mathbf{E}^0)$ |

**Training**: Algorithm 2 gives the procedure for a full training step, including, the forward and reverse diffusion. For some randomly sampled $t$ in the noising chain and Gaussian noise $\epsilon$ (line 1), we add $\epsilon$ to the clean data with a weight determined by the schedule at $t$ (line 2). For each possible edge location in the adjacency matrix, we choose the distribution from the transition matrix corresponding to its edge type and sample from it to determine the updated edge type (line 3). We compute extra

per-node and per-graph features encoding structural properties of the newly formed graph to help with the model's prediction (line 4) and run the forward pass (line 5). Finally, we optimize the loss (line 6), which is a (weighted) sum of the mean square error between the clean and predicted node data, and cross entropy between the corresponding edge classes.

**Sampling**: Algorithm 3 describes the sampling. We start from sampled Gaussian noise at the nodes and adjacency entries sampled from the prior distribution of edge types within clean graphs (line 1). Then, for each time step in the reverse chain (line 2), we compute the current structural features, and have the model predict the clean graph (line 4), as during training. Subsequently, we sample the necessary noise to partially renoise the model's current guess for the node data (lines 5 and 6).

For the predicted probabilities of edges we also apply partial renoising, albeit by manipulating the probability of each state at each location via the transition matrices, before sampling a new discrete outcome in each location for the outcome distribution (lines 7 and 8).

## 4 EVALUATION

**Baselines**: Since DLGrapher is the first work investigating attributed graph generation with complex node features, we propose a couple of different baselines that combine the best-in-class generators for node features, i.e., TVAE (Xu et al., 2019) (VAE-based) or TabDDPM (Kotelnikov et al., 2023) (diffusion-based), with a state-of-the-art graph synthesizer, i.e., DiGress Vignac et al. (2023). Based on preliminary experiments, we set DLGrapher's sVAE compression factor $f' = \lfloor \frac{f}{4} \rfloor$ in all experiments for best trade-off between compression and quality. We further test DLGrapher without sVAE and structure coarsening, termed Dual Diffusion in the following. Here, we compare against the state-of-the-art methods of DiGress (Vignac et al., 2023) and GruM (Jo et al., 2024).

**Metrics**: Alongside compute time, our main results consider the quality of graph topology, node features, and the interaction between the two. To measure structure quality we monitor four graph metrics and compute the Maximum Mean Distance (MMD) between the distribution of their values over the synthetic and the real graphs. Specifically, following prior studies (Martinkus et al., 2022; Vignac et al., 2023; Jo et al., 2024), we choose as metrics: the distribution of node degrees (Deg), the eigenvalues of the normalized graph Laplacian (Spec), clustering coefficients (Clus), and orbit counts (Orb). To evaluate node features in isolation, we treat nodes as tabular data rows and apply standard metrics checking the distance between column shapes (Shape) and pairwise correlations (Pair Trend) in synthetic and real samples Patki et al. (2016). To examine relationships between graph structure and node features we choose a binary-valued node feature and compute the MMD selectively on the node neighbors with the label set. Additionally, we test downstream utility of ML tasks via accuracy metric of node classification when using the same binary node feature as target. For molecular data, we match other works Vignac et al. (2023); Jo et al. (2024) and focus on assessing utility by measuring the ratio of valid/unique/novel synthesized molecules.

**Datasets**: We employ three public datasets describing multi-feature entities and their relationships for experiments on larger graphs with complex node features, plus a benchmark dataset for experiments on smaller graphs with simple node features. Specifically, the former comprises two social network datasets, Twitch (Rozemberczki & Sarkar, 2021) and Event (Allan Carroll, 2013), with complex node features. Here, we harness as target binary label for downstream ML tasks and MMD, respectively, whether a user may earn money from the platform and whether the gender of a user is marked as female. The third dataset is OGBN-arxiv (Hu et al., 2020), a citation network where articles, i.e. the nodes, are assigned a 128 dimensional embedding of the title and abstract, i.e. the node features. We interpret the node embeddings as numerical columns and use a binary target label of whether a paper is registered to one of the top four most popular categories. Since all three datasets entail a single huge graph, we use random walks to create a set of smaller graphs with a configurable number of nodes for learning and evaluation. We use either small graphs of 160 nodes or large graphs of 260 nodes. Finally, the benchmark datasets is QM9 (Wu et al., 2017) comprising graphs representing small molecules of up to 9 nodes with categorical node and edge features.

### 4.1 COMPLEX NODE DATA

Table 1 showcases the performance comparison for complex-node graphs. We observe that both versions of DLGrapher, Dual Diffusion significantly outperform the baselines on mixed structure-

| Dataset/Method | MMD (↓) | | | | Column (↑) | | Tgt. Col. MMD (↓) | Downstr. Util. (↑) | Epoch time s (↓) | | # Train Nodes |
| | Deg | Spec | Clus | Orb | Shape | Pair Trend | | | Train | Sample | |
|---|---|---|---|---|---|---|---|---|---|---|---|
| **Twitch** | | | | | | | | | | | |
| DiGress+TVAE | .344 | .039 | .257 | .124 | .867 | .913 | .281 | .0 | 5.54 | 742 | 160 |
| DiGress+TabDDPM | .317 | .036 | .240 | .215 | .907 | **.971** | .323 | .0 | 5.54 | 741 | 160 |
| *Dual Diffusion* | **.010** | **.009** | **.060** | **.055** | **.945** | .957 | **.002** | **.796** | 5.57 | 748 | 160 |
| *DLGrapher* | .049 | .038 | .176 | .056 | .866 | .930 | .024 | .685 | **2.11** | **294** | 94.93 (-40.66%) |
| **Event** | | | | | | | | | | | |
| DiGress+TVAE | .307 | .073 | .280 | .491 | **.952** | .793 | .202 | .0 | 5.54 | 741 | 160 |
| DiGress+TabDDPM | .194 | .194 | .263 | .395 | .835 | .710 | .101 | .580 | 5.55 | 742 | 160 |
| *Dual Diffusion* | **.005** | **.007** | .196 | .077 | .821 | **.823** | **.002** | **.642** | 5.56 | 748 | 160 |
| *DLGrapher* | .014 | .030 | **.157** | **.036** | .760 | .567 | .004 | .616 | **2.16** | **305** | 94.42 (-40.98%) |
| **OGBN-arxiv** | | | | | | | | | | | |
| DiGress+TVAE | .042 | .032 | >1 | .413 | **.946** | **.975** | .016 | **.777** | 9.16 | 1272 | 160 |
| DiGress+TabDDPM | .039 | .032 | .967 | .385 | .500 | .529 | .046 | .469 | 9.17 | 1272 | 160 |
| *Dual Diffusion* | **.002** | **.006** | **.116** | **.082** | .874 | .966 | **.002** | .703 | 9.19 | 1280 | 160 |
| *DLGrapher* | .015 | .035 | .183 | .155 | .607 | .752 | .009 | .741 | **3.59** | **479** | 94.21 (-41.11%) |

Table 1: Main result on complex attributed graphs: showing the advantage in higher quality of graph structure, feature, their interaction, downstream tasks, and training time per epoch.

| Dataset | MMD (↓) | | | | Column (↑) | | Tgt. Col. MMD (↓) | Downstr. Util. (↑) | Epoch time s (↓) | | # Train Nodes |
| | Deg | Spec | Clus | Orb | Shape | Pair Trend | | | Train | Sample | |
|---|---|---|---|---|---|---|---|---|---|---|---|
| Twitch large | .020 | .017 | .177 | .050 | .858 | .940 | .009 | .727 | 5.56 | 784 | 155.09 (-40.35%) |
| Event large | .006 | .020 | .162 | .075 | .768 | .520 | .001 | .599 | 5.59 | 826 | 153.53 (-40.95%) |
| OGBN-arxiv large | .010 | .025 | .441 | .071 | .561 | .623 | .002 | .706 | 9.30 | 1277 | 153.76 (-40.86%) |

Table 2: Results for larger variants of complex graphs.

feature metrics by 12.9 on the target column MMD and 25.2% on downstream utility. For each of the three datasets, we create a train/test/evaluation split from 200 graphs with 160 nodes each. We underline that this size is close to what existing works on attributed graphs are able to synthesize. For structure metrics, we find that our versions of DLGraphertend to significantly outperform baselines using structure-only diffusion. This suggests that incorporating node features into the diffusion model also helps better model the edge connectivity. Meanwhile, baselines aided by tabular synthesizers do better on node feature metrics, sometimes outperforming our proposed method. Another noteworthy observation is that DLGrapher can better preserve column correlation (see Pair Trend) under the applied graph coarsening ratio in Event; and, the overall poor performance of the TabDDPM-aided DiGress generating high-quality word embeddings for OGBN-arxiv. Dual Diffusion without any coarsening is clearly best in accounting for structure and node features together, with DLGrapher always being a close second. Indeed, DLGrapher's latent embedding reduces node counts by > 40% in all cases, leading to approximately 2.5 times faster training and sampling epoch times trading a small quality loss for speed. Our latent dual diffusion has consistently the same performance gains across all datasets.

## 4.2 SCALABILITY

In Table 2, we further test the scalability of DLGrapher specifically using large 260-node graphs with complex node features. Comparing the results against Table 1 shows that DLGrapher scales well, obtaining the same performance on larger graphs as on smaller graphs. All while requiring approximately the same run time as other baselines require for the smaller graphs. We exclude baselines due to prohibitive runtimes on larger graphs. For DLGrapher, train epoch times are only marginally higher, while sampling increases with the number of synthesized nodes. The node coarsening rate also remains very similar to previous tests, showing that the structure coarsening can reliably reduce the size by at least 40% across various graphs of various sizes.

### 4.3 SIMPLE NODE DATA

Finally, we evaluate DLGrapher on the QM9 molecules benchmark dataset to showcase that DL-Grapheris competitive even small graphs with simple categorical node features. For DiGress and GruM, we report their scores from Jo et al. (2024). Note that the DiGress's original paper reports a marginally higher mean percentage of valid molecules of 99% but a lower percentage of unique molecules out of the valid ones of 96.2%. For both reported results, the relative ranking of DiGress remains unchanged.

DLGrapher achieves similar performance on the ratio of valid molecules and unique molecules out of the valid ones, coming in second for both metrics. Furthermore, regarding the ratio of novel molecules not present in the training set over out of the valid and unique ones, DLGrapheris the best. We attribute this to the continuous diffusion component on the node features, increasing the diversity within the overall diffusion process. The rel-

| Dataset/Method | Utility % ($\uparrow$) | | |
|---|---|---|---|
| | Valid | Unique/Valid | Novel/Valid-Unique |
| QM9 | | | |
| DiGress | 98.19 | 96.67 | 25.58 |
| GruM | **99.69** | **96.90** | 24.15 |
| *Dual Diffusion* | 99.46 | 96.82 | **36.10** |

Table 3: Results on smaller molecular data graphs with categorical node and edge features.

atively low number of novel graphs across the board is due to graphs in QM9 having at most 9 nodes and a relatively large train set.

## 5 CONCLUSION

Attributed graphs with rich node features are a critical data type in applications across multiple domains such as social networks, financial transactions or molecular biology. The prior art, unfortunately, is limited to synthesizing only single attributed or small graphs. In this paper, we present DLGrapher, a dual latent diffusion model for attributed graphs - modeling the graph structure and node feature as a joint discrete and continuous diffusion process. We first represent the complex node feature as an embedding through a structure-aware VAE. We then apply a reversible coarsening scheme to find a structure embedding in the original graph space, i.e., virtual nodes and virtual edges through aggregating nodes and edges. The dual diffusion model then trains noise-predicting networks that can denoise the continuous feature embedding of virtual nodes and the discrete virtual edges. Our evaluations on small and large attributed graphs show that DLGrapher captures node feature and edge interdependencies 12.9x better and improves performance on downstream tasks by 25.2%.

## 6 ETHICS AND REPRODUCIBILITY STATEMENT

**Ethics**: Our proposed graph generative models have broad applications in modeling molecular structures used in drug discovery or material science applications and human interactions on social media, professional networks, or social contagion situations. As a generative model, our solution can help improve productivity (e.g., propose plausible new drug candidates for further validation) and alleviate the need for third parties to directly tap into confidential or privacy-sensitive data when answering questions about it (e.g., finding out how some disease spreads amongst different user groups).

**Reproducibility**: To ensure the reproducibility of our research, we include the code for the proposed model and datasets as supplementary material.

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

## A  LEARNING SETUP

The following describes our procedure of training a synthesizer that harnesses diffusion backbone alongside latent embedding mechanism. Before training the diffusion model, we first pretrain the sVAE used to reduce feature dimensionality, then apply the full latent transformation to the train data in preparation. Thus, the diffusion loss is optimized directly in the reduced compression space, and the mapping to the original space is only performed when a complete output is required, like during evaluation. Doing so, we avoid involving the decompression during training, as to not increase training cost. Consequently, we also keep the calculation of extra node and graph spectral features in the compressed space, as their aim is to help the model understand the structural properties of the partially noisy graph at any given time step. On another note, the structure and node feature components of the embedding mechanisms can also be applied independently and are effectively a pre/post-processing step on top of the main diffusion network. As such, they are also compatible with any graph generation model that allows for attributed nodes in the case of feature compression or edges in the case of structure compression.

## B  NOTATION

Table 4 recaps the notation used throughout the manuscript to describe the graph representations at different stages in the framework, and the components making up the latent embedding and dual diffusion.

| Notation | Description |
|---|---|
| $\phi_\theta$ | denoising model $\phi$ parameterized by $\theta$ |
| $G = (\mathbf{V} \in \mathbb{R}^{n^* \times f^*}, \mathbf{M} \in \{0,1\}^{n^* \times n^*})$ | original attributed graph |
| $\mathbf{Z} \in \mathbb{R}^{n^* \times f}$ | latent node feature embedding |
| $G_0 = (\mathbf{X} \in \mathbb{R}^{n \times 2f}, \mathbf{E} \in \mathbb{N}_{15}^{n \times})$ | embedded attributed graph |
| $G^t / \mathbf{X}^t / \mathbf{E}^t$ | graph/nodes/edges after $t$ noise steps |
| $\mathbf{x}_i$ & $e_{ij}$ | node embedding $i$ and edge value $i, j$ |
| $d_\theta$ & $e_\theta$ | node feature VAE decoder & encoder |
| $q(G^t \mid G^t)$ | probability distribution of $G^t$ given $G^t$ |
| $\mathbf{Q}^t$ | edge-type transition matrix at noise step $t$ |
| $\overline{\mathbf{Q}}^t$ | edge-type transition matrix for noise steps up to $t$ |
| $p_E$ | likelihood of each state for all possible edges in $\mathbf{E}$ |
| $q_E$ | prior probability for each edge type in $\mathbf{E}$ |
| $\alpha^t$ & $\sigma^t$ | parameters for noise strength schedule up at step $t$ |
| $\bar{\alpha}^t$ & $\bar{\sigma}^t$ | parameters for noise strength schedule up to step $t$ |
| $\alpha^{t \to t-1}$ & $\sigma^{t \to t-1}$ | parameters for noise strength at step $t$ given $\mathbf{X}^0$ & $\mathbf{X}^t$ |
| $\epsilon^X$ | sampled noise for corrupting nodes |

Table 4: Overview of the main notation used in the main text and its description.

## C  STRUCTURE DECOARSENING DETAILS

Algorithm 4 provides more details on the structure decoarsening, which reverses the steps of the coarsening. We first split back each node representation in two (line 1) and add the edges between nodes previously in the same pair (lines 2 to 3). We subsequently expand each edge in the compressed graph to the original graph edges it aggregates, adding them to the new graph structure (lines 4 and 5). Finally, we remove any dummy zero-filled nodes, reindexing the graph to account for any reduction in nodes (line 6), and performing a forward pass through the decoder to restore the original state space of nodes (line 7).

## D  SVAE ARCHITECTURE

Figure 4 visualizes the architecture of sVAE for the case of two encoding and decoding layers, respectively. Each layer takes as input a representation of the node features after the previous step,

**Algorithm 4** structure Decoarsening

**Input**: feature decoder $d_\theta$,
$G = (\mathbf{X} \in \mathbb{R}^{n' \times 2f'}, \mathbf{E} \in \mathbb{N}_{15}^{n' \times n'})$
1: $\mathbf{Z} \leftarrow UnpairNodes(\mathbf{X})$
2: **for** $i \leftarrow 1, 3, ..., 2n' - 1$ **do**
3:      $\mathbf{M}_{i,i+1} \leftarrow \mathbf{M}_{i+1,i} \leftarrow 1$
4: **for** $(i, j) \leftarrow DecEdges(\mathbf{E})$ **do**
5:      $\mathbf{M}_{i,j} \leftarrow \mathbf{M}_{j,i} \leftarrow 1$
6: $\mathbf{Z}, \mathbf{M} \leftarrow RemoveZeroNodes(\mathbf{Z}, \mathbf{M})$
7: $\hat{\mathbf{V}} \leftarrow d_\theta(\mathbf{Z}, \mathbf{M})$
8: **return** $\hat{\mathbf{V}}, \mathbf{M}$

along with the connectivity information of the graph. As is typical in VAEs, the encoder estimates the parameters of a prior distribution, which, in our case, are the mean and variance of a Gaussian. Consequently, the decoder expects a sample drawn from the latent distribution as input. Finally, we use a different activation function for each feature based on whether it represents a value for tabular numerical feature or is part of a one-hot embedding for a tabular categorical feature. For node features that do not originally encode a tabular data row, we consider each feature to be a unique numerical column.

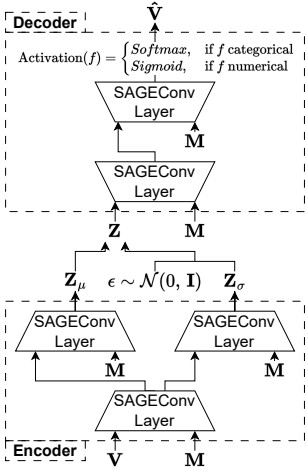

Figure 4: sVAE architecture with 2 encoder and decoder layers each.

## E   SYNTHETIC COMPLEX NODE DATA SAMPLES

Table 5 showcases an example graph for each tested method and the Twitch and Event datasets.

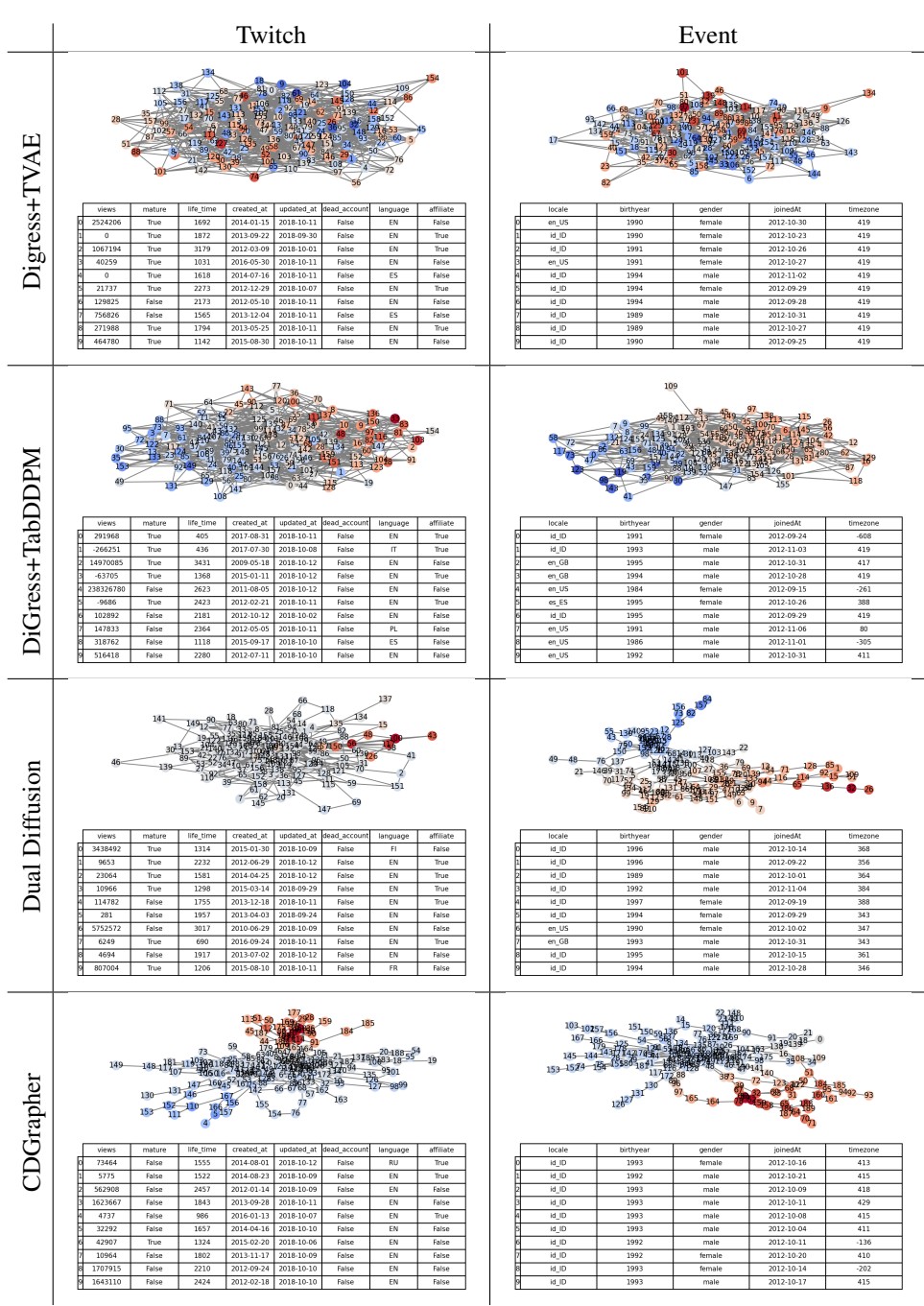

Table 5: Samples from the Twitch and Event datasets generated by the two baselines (DiGress + TVAE, DiGress + TabDDPM) and our two proposed methods (Dual Diffusion without feature nor structure compression, Twitch). For readability we only show the node feature values of the first 10 nodes.

