# OpenReview forum: "DLGrapher: Dual Latent Diffusion for Attributed Graph Generation"
_ICLR.cc/2025/Conference — ICLR 2025 Conference Withdrawn Submission_

### Official Review · Reviewer_pPC4 · 2024-10-27

**Soundness:** 2
**Presentation:** 2
**Contribution:** 2
**Rating:** 3
**Confidence:** 4

**Summary:**

This paper presents a DUAL LATENT DIFFUSION model for graph generation. For continuous node features and discrete structural features, the authors use continuous diffusion and discrete diffusion models to generate the corresponding features, respectively. Additionally, the authors propose a graph coarsening method to reduce the burden of diffusion in generating graphs, making large-scale graph generation feasible.

**Strengths:**

1. Considering a generative model applicable to large-scale graphs is an important research direction in the field of graph generation.
2. The authors design dual diffusion from the perspectives of continuous node features and discrete structural features, representing a novel exploration in graph generation.

**Weaknesses:**

1. The motivation for the model design is not clearly expressed. For instance, why is it necessary to obtain structure-aware features through coarsening?
2. The description of the model is unclear. For example, how does the decoarsening process restore the graph?
3. From Table 1, dual diffusion performs better in most cases, indicating that the decoarsening process has a significant impact on the generation results.
4. More large-scale molecular generation datasets need to be compared, such as MOSES and GuacaMol.

**Questions:**

1. How is the number of virtual nodes determined after coarsening? I.e., what is the number of nodes at the start of sampling?
2. Why are the results of using only DiGress not included in Table 1? The results of GruM should also be included.

---

### Official Review · Reviewer_ijjH · 2024-11-04

**Soundness:** 2
**Presentation:** 3
**Contribution:** 2
**Rating:** 6
**Confidence:** 4

**Summary:**

The paper proposes a latent generative framework for graph generation. The method uses two auto-encoder, one encode node features then one encodes structural features. In the latent space obtained from the two auto-encoders, the authors employs a dual diffusion like the one in GDSS to jointly model the node and edge, where node is represented by continuous value and edge is represented by discrete value. Experiment shows the proposed method outperform baselines in terms of quality and efficiency.

**Strengths:**

1. The paper is well-written and I can easily see the contribution of the papers.

2. Coarsening is used in the latent space to reduce modeling complexity, which is intuitive and convincing.

**Weaknesses:**

1. It is not clear how the nodes are dynamically changed during the diffusion/denoising

2. Typo in figure 1: iv->vi, and the equation at (4) is confusing.

3. The author need to differentiate its work against GDSS, which also uses a dual diffusion for modeling graph. Moreover, latent graph diffusion work are missing [1]

4. Can you provide a theoretical and empirical time analysis on the proposed method. I feels that theoretically the time complexity is still O(N^2). However, It would be great to see a sensitivity analysis of wall-clock time vs. #nodes.

[1] Chen, Xiaohui, et al. "Nvdiff: Graph generation through the diffusion of node vectors." arXiv preprint arXiv:2211.10794 (2022).

**Questions:**

See weakness.

---

### Official Review · Reviewer_rHRm · 2024-11-04

**Soundness:** 1
**Presentation:** 2
**Contribution:** 1
**Rating:** 3
**Confidence:** 3

**Summary:**

This paper presents DLGrapher for generating large graph structures with high-dimensional node features. Specifically, the proposed framework consists of three components: 1) graph variational autoencoder to encode the node features in a low-dimensional embedding space, 2) reversible coarsening scheme to compress the large graph structures into a low dimension graph, and 3) Dual diffusion model which simultaneously generates the latent embeddings of nodes and edge type by leveraging the DDPM [1] for node features and D3PM [2, 3] for edge types. The authors evaluate DLGrapher on social network generation tasks and molecular graph generation tasks, showing that the proposed method is efficient in reducing the training and sampling time.


[1] Ho et al., "Denoising Diffusion Probabilistic Models", NeurIPS 2020.

[2] Austin et al., "Structured Denoising Diffusion Models in Discrete State-Spaces", NeurIPS 2021.

[3] Vignac et al., "DiGress: Discrete Denoising diffusion for graph generation", ICLR 2023.

**Strengths:**

- The proposed method is tailored for generating a large graph by efficiently compressing the node features and graph structures.
- The authors achieved the performance improvement by generating the node features on the latent space.

**Weaknesses:**

- The evaluation setting is limited to several datasets and baselines. Specifically, the authors do not evaluate it on a well-studied benchmark such as SBM, Planar, Cora, etc. Also, the authors include only a few baselines: variants of DiGress, and GruM. Therefore, it is hard to clearly demonstrate the effectiveness of the proposed method.

- A relevant work [4] is not compared. I highly recommend that the authors experimentally and conceptually compare this work, as it aims to generate large-scale graphs.

- Even though the proposed coarsening scheme is efficient, it is limited to a single coarsening step, hindering a further application for larger graphs.

[4] Chen et al., "Efficient and Degree-Guided Graph Generation via Discrete Diffusion Modeling", ICML 2023.

**Questions:**

- In Table 3, does "Dual Diffusion" mean the DLGGrapher with the graph coarsening?

- To my understanding, the virtual nodes play a pivotal role in coarsening the graphs. Can the virtual nodes actually embed structural information?

---

### Official Review · Reviewer_8fkb · 2024-11-05

**Soundness:** 3
**Presentation:** 1
**Contribution:** 2
**Rating:** 3
**Confidence:** 4

**Summary:**

The paper presents a diffusion-based generative model for graphs, incorporating a discrete diffusion model for generating graph structure and a continuous diffusion model for node features. In comparison to existing literature, the approach includes two distinct design elements: the application of dual diffusion models and the construction of the diffusion process on a compressed view of the graphs. With these design elements, the proposed model demonstrates improved performance over the selected baselines in terms of the similarity of generated graphs and node features, while also achieving lower training and sampling times.

**Strengths:**

The paper’s strengths include its benchmarking of attributed graph generation using diffusion models. The use of a compressed graph as the diffusion latent is also a novel approach. Additionally, integrating both a discrete diffusion model and a continuous diffusion model adds further novelty to the proposed method.

**Weaknesses:**

1. While constructing the diffusion process on a "compressed graph" is interesting and novel, the paper lacks sufficient rationale or justification for this modeling choice. I would expect to see a discussion on why a compressed graph was chosen as the latent variable for the diffusion process, rather than numerical tensors, which are used for this purpose in [1,2].

2. The integration of a discrete graph diffusion model with a continuous one lacks fundamental innovation, appearing to be a straightforward approach to address attributed graph generation. The authors might consider detailing any technical challenges they encountered and how these were addressed.

3. The proposed graph coarsening method does not significantly reduce the computational complexity, as the order of complexity remains unchanged when training and sampling from the diffusion model on the original, uncompressed graph.

4. The presentation of the reversible structure compression is unclear and difficult to follow, as many notations in Algorithm 1 are introduced without proper definition. Please refer to my questions for clarification.

5. Given that softmax and sigmoid functions are applied to the output layer of the structure-aware VAE, mean square loss may not be the most suitable choice for training. Cross-entropy loss would likely be more appropriate.

6. The results in Tables 1 and 2 are reported without standard deviations, making it difficult to assess the significance of the improvements. Additionally, Table 2 lacks any baseline comparisons.

7. Some recent diffusion-based models specifically designed for generating large graphs, such as [3,4], should be included as additional baselines.


**References**

[1] Cai Z, Wang X, Zhang M. Latent Graph Diffusion: A Unified Framework for Generation and Prediction on Graphs[J]. arXiv preprint arXiv:2402.02518, 2024.

[2] L. Yang et al., "Graphusion: Latent Diffusion for Graph Generation," in IEEE Transactions on Knowledge and Data Engineering, vol. 36, no. 11, pp. 6358-6369, Nov. 2024, doi: 10.1109/TKDE.2024.3389783.

[3] Chen, Xiaohui, et al. "Efficient and Degree-Guided Graph Generation via Discrete Diffusion Modeling." International Conference on Machine Learning. PMLR, 2023.

[4] Trivedi, Puja, et al. "Editing Partially Observable Networks via Graph Diffusion Models." Forty-first International Conference on Machine Learning.

**Questions:**

1. The following questions pertain to the notations in Algorithm 1:

-  What do $i,j$ represent when used to index $\mathbf{z}$ and "Pair"?
- What does "$\mathrm{Pair}_i$" refer to?
- What is "$\mathrm{max}(\mathrm{Pair})$"?
- How was $\mathrm{X}$ initialized before appending all the concatenated node features?

2. What criterion is used to identify trailing zeros in the node features, given that these features are generated by the continuous diffusion model?

3. Considering the description of the QM9 dataset as "small graphs with simple categorical node features", could the authors provide some insights into why continuous modeling of node features might result in a higher novelty score, given the discrete nature of the data? Additionally, if this is the case, why might the continuous GruM model not exhibit a similarly high novelty score?

---

### Official Review · Reviewer_MuwG · 2024-11-06

**Soundness:** 2
**Presentation:** 2
**Contribution:** 2
**Rating:** 3
**Confidence:** 3

**Summary:**

The paper introduces a dual latent diffusion framework that capture the interaction between node feature and graph structure.

**Strengths:**

1.The paper studies the generation of large and complex attributed graphs.
2.The proposed method considers both discrete latent structures and continuous latent features.
3.The author evaluates the model's effectiveness across multiple metrics.

**Weaknesses:**

1. Most generative graphs study molecular generation. The generation model in the social network is not particularly important and the motivation for generating social graphs should be clarified.
2. The authors only provide a baseline model with two variants. The baseline models provided appear to be insufficient to support the conclusions.
3. The introduction of the methods seems repetitive. I suggest making this section clearer by removing unnecessary information.
4. The proposed dual latent diffusion method combining feature latent and edge latent is straightforward.
5. Although the performance on QM9 is impressive, this graph appears to be largely resolved.

**Questions:**

My concerns are as follows:
1.What is the motivation for using a generative model in the social network?
2.Could you provide more insights or an ablation study into dual diffusion design?
3.How does the weight of cross entropy loss for the edge types change affect the performance?

---

### Note · Authors · 2024-11-18

**Comment:**

We thank all the reviewers for their valuable feedback. We plan to address the raised points for a revised version of the work and, as such, withdraw the current paper.

**Withdrawal Confirmation:**

I have read and agree with the venue's withdrawal policy on behalf of myself and my co-authors.